# Evaluation of the Efficiency of Random and Diblock Methacrylate-Based Amphiphilic Cationic Polymers against Major Bacterial Pathogens Associated with Cystic Fibrosis

**DOI:** 10.3390/antibiotics12010120

**Published:** 2023-01-08

**Authors:** Magali Casanova, Hamza Olleik, Slim Hdiouech, Clarisse Roblin, Jean-François Cavalier, Vanessa Point, Katy Jeannot, Baptiste Caron, Josette Perrier, Siméon Charriau, Mickael Lafond, Yohann Guillaneuf, Stéphane Canaan, Catherine Lefay, Marc Maresca

**Affiliations:** 1CNRS, Aix-Marseille Univ, LISM UMR7255, IMM FR3479, 13402 Marseille, France; 2Aix Marseille Univ, CNRS, Centrale Marseille, iSm2 (UMR7313), 13013 Marseille, France; 3Aix Marseille Univ, CNRS, ICR, UMR7273, 13397 Marseille, France; 4Laboratoire Associé au CNR de la Résistance aux Antibiotiques, Centre Hospitalier Universitaire de Besançon, Boulevard Fleming, 25000 Besançon, France

**Keywords:** radical polymerization, antibiotic resistance, antimicrobial, antibacterial, amphiphilic cationic polymers, cystic fibrosis, *Mycobacterium abscessus*, *Pseudomonas aeruginosa*, *Staphylococcus aureus*

## Abstract

Cystic fibrosis (CF) is associated with repeated lung bacterial infection, mainly by *Pseudomonas aeruginosa*, *Staphylococcus aureus*, and *Mycobacterium abscessus*, all known to be or becoming resistant to several antibiotics, often leading to therapeutic failure and death. In this context, antimicrobial peptides and antimicrobial polymers active against resistant strains and less prompt to cause resistance, appear as a good alternative to conventional antibiotics. In the present study, methacrylate-based copolymers obtained by radical chemistry were evaluated against CF-associated bacterial strains. Results showed that the type (Random *versus* Diblock) and the size of the copolymers affected their antibacterial activity and toxicity. Among the different copolymers tested, four (i.e., Random_10200_, Random_15000_, Random_23900_, and Diblock_9500_) were identified as the most active and the safest molecules and were further investigated. Data showed that they inserted into bacterial lipids, leading to a rapid membranolytic effect and killing of the bacterial. In relation with their fast bactericidal action and conversely to conventional antibiotics, those copolymers did not induce a resistance and remained active against antibiotic-resistant strains. Finally, the selected copolymers possessed a preventive effect on biofilm formation, although not exhibiting disruptive activity. Overall, the present study demonstrates that methacrylate-based copolymers are an interesting alternative to conventional antibiotics in the treatment of CF-associated bacterial infection.

## 1. Introduction

Cystic fibrosis (CF) is a recessive genetic disease affecting between 70,000 and 100,000 people worldwide [1], with an average incidence between 1/3000 and 1/6000 in populations of European descent [2,3]. Although complications are possible in nearly all organs, respiratory disease is the most severe symptom, and the most frequent cause of death [4]. For most of CF patients, the thickening of the airway mucus leads to recurrent and/or persistent colonization of the lung by microorganisms (pathogenic or opportunistic) eventually leading to airway inflammation, respiratory failure, and death [5]. Although many pathogenic microorganisms can colonize the lung of CF patients, *Pseudomonas aeruginosa* and *Staphylococcus aureus* are the most frequently found bacteria [6,7]. The prevalence of *P. aeruginosa* increases with age, ranging from ~25% in children and up to 70–80% in adult CF patients [8]. This latter pathogen is known for its high level of antibiotic resistance related to its high rate of mutation. That allows it to rapidly evolve and adapt to a multitude of conditions and drugs, resulting in recalcitrant and relapsing infections [9,10]. Approximately 50% of adult CF patients are infected by *S. aureus* but, in contrast to *P. aeruginosa*, its prevalence is higher in children, exceeding 60% [8]. Importantly, the prevalence of *S. aureus* infection in the adult CF patients increased steadily during the last decades, whereas the prevalence of *P. aeruginosa* tends to be constant or to decrease [8]. More alarming is the increasing diffusion in CF patients of antibiotic-resistant strains of *S. aureus*, mainly methicillin-resistant *S. aureus* (MRSA) strains [11]. Last, with the increase in the life span of CF patients, in addition to *S. aureus* and *P. aeruginosa*, new pathogens have now become clinically relevant. Nontuberculous mycobacteria (NTM) are among such opportunistic pathogens of which *Mycobacterium abscessus* represents the most significant example [12], with a prevalence ranging from 2% to 28% [8]. Infections by NTM, and notably *M. abscessus*, represent a serious problem for CF patients. *M. abscessus* exists as two distinct colony morphotypes, a smooth (S) variant and a rough (R) one [13], which may evolve differently in infected patients [14]. The S variant is responsible for the primo infection, while the R variant is often associated with late stage severe pulmonary infection [15,16]. The main difference between the two variants is the presence of surface-associated glycopeptidolipids (GPL) in the S form, which are absent in the R one [13]. Moreover, *M. abscessus* is intrinsically resistant not only to nearly all conventional anti-tuberculosis drugs, but also to most currently available antibiotics. This makes infections due to this mycobacterium, referred to as an “antibiotic nightmare” [17], very difficult to treat and eradicate [18,19,20]. Furthermore, postoperative severe complications are often observed in patients with *M. abscessus* infection prior to lung transplantation [21]. With nearly ⅓ of adult CF patients co-infected by *P. aeruginosa* and *S. aureus* [22,23], and since *M. abscessus* may also be present in these patients, the development of novel efficient drug treatments active against each of these strains is urgently needed. However, due to their intrinsic and/or acquired resistance to already available antibiotics, the chances for a drug to be active on the three bacterial species at the same time are very limited [24]. The development of alternative antimicrobial agents with a large spectrum able to act through different mechanisms compared to conventional antibiotics, can thus offer new therapeutic options and thwart current multidrug resistance. In this context, naturally occurring antimicrobial peptides (AMPs), found in all living organisms, show particular interest either for administration as a monotherapy or combined with other drugs [25,26]. Contrary to conventional antibiotics that classically target proteins and enzymes, most of the AMPs act on the bacterial membrane [25,26]. The original mechanism of action of AMPs differing from the one of conventional antibiotics allows them to remain active against bacteria resistant to antibiotics and to be less prompt at causing resistance [27,28]. However, although some AMPs succeeded in clinical trials, their cost/time of production, lability, and possible cytotoxicity, have prevented the development of systemic applications of AMPs-based therapeutics [29]. In this context, different molecules inspired from AMPs have been designed in order to maintain the benefits and erase the limitations of AMPs [30,31]. Most of these synthetic molecules, either based on a peptidic or non-peptidic scaffold, have been designed to mimic the cationic and amphiphilic properties of native AMPs, which is the key determinant of their antibacterial activity [31,32]. Among them, non-peptidic cationic amphiphilic polymers recently received growing attention as alternative antimicrobial agents [29,33,34,35,36,37,38,39]. Similar to AMPs, cationic antimicrobial polymers are active against different bacterial species including the ones already resistant to conventional antibiotics and do not seem to cause resistance [29,40,41,42,43,44,45,46,47]. In addition, these polymers also exhibit some inherent advantages over AMPs, such as easy and low cost/fast production allowing large scale synthesis (in kilograms), versatile chemical structures, unique self-assembly properties, low cytotoxicity, prolonged antimicrobial activity, and stability to protease degradation [29,47,48]. In the present study, different methacrylate-based cationic copolymers were synthetized by a radical polymerization using dimethylaminoethyl methacrylate (DMAEMA) as hydrophilic monomer and butyl methacrylate (BMA) as hydrophobic monomer (Figure 1). More precisely, Diblock copolymers named poly(BMA)-*b*-poly(DMAEMA)(PBMA-*b*-PDMAEMA), and Random copolymers (PBMA-*co*-DMAEMA), from 2800 to 23,900 g/mol, were synthetized.

Those Random and Diblock copolymers of different sizes were tested for their antimicrobial activity against the three major bacteria found in CF patients, i.e., *P. aeruginosa*, *S. aureus*, and *M. abscessus*. Their innocuity and their mechanism of action were also investigated, allowing for the identification of four promising candidates for the treatment of infections found in CF patient.

## 2. Results

### 2.1. Antimicrobial Activity of the Copolymers

The characteristics of the Random and Diblock copolymers, i.e., molecular weight and *F*_DMAEMA_ corresponding to the DMAEMA molar ratio in the final copolymers, are given in Table 1.

The bacterial activity of the Random and Diblock copolymers of different sizes was evaluated through determination of their minimum inhibitory concentrations (MIC). The comparison of the antibacterial activity of the different copolymers shows that their activity has no correlation with their *F*_DMAEMA_. As an example, Random_10200_, Diblock_18400_, and Diblock_20500_, all having a *F*_DMAEMA_ of 0.64, display different MIC values (Table 2); this is also true for Random_9300_, Random_14600_, and Diblock_6800_ or with Diblock_9500_ and Diblock_17900_ sharing the same *F*_DMAEMA_ values but giving different MIC (Table 2). Results demonstrated that copolymer activity rather relies on their type (i.e., Random or Diblock) and their size/molecular weight. Regarding Random copolymers (Table 2), MIC values on *M. abscessus* S range from 6.25 to 25 µg/mL with a tendency of decreased/weaker activity correlated to increasing sizes. For example, the most active Random copolymer on *M. abscessus* S (i.e., the one with the lowest MIC value) is Random_2800_ with a MIC of 6.25 µg/mL, whereas Random copolymers with a size ranging from 14,600 to 23,900 g/mol possess MIC of 25 µg/mL. Oppositely, for *P. aeruginosa* and *S. aureus*, Random copolymers display a tendency of increased/improved activity when their size increases (MIC ranging from 12.5 to 400 µg/mL and from 12.5 to 200 µg/mL for *P. aeruginosa* and *S. aureus*, respectively). Indeed, the MIC values of Random_2800_ on *P. aeruginosa* and *S. aureus* are 200 µg/mL, whereas Random_23900_ possesses a MIC of 12.5 µg/mL on both strains.

Regarding Diblock copolymers (Table 2), as for Random ones, the MIC values on *M. abscessus* S also display decreased activity with increase in their size (from 25 to 100 µg/mL for Diblock_6800_ and Diblock_21900_, respectively). In the case of *P. aeruginosa* and *S. aureus*, the size of the Diblock copolymers has little influence on the MIC values, ranging from 50 to 100 µg/mL and from 12.5 to 25 µg/mL, respectively.

Finally, it should be mentioned that none of the tested Random and Diblock copolymers possesses antibacterial effect on *M. abscessus* R (MIC >400 µg/mL).

### 2.2. Innocuity of the Copolymers

The cytotoxic activity of the copolymers was measured after 48 h exposure of non-tumorigenic human lung epithelial cells (BEAS-2B cells) to increasing concentrations of Random or Diblock copolymers (Figure 2). This assay allowed the determination of the IC_50_ (Inhibitory Concentration 50%), corresponding to the concentration of compound that inhibits 50% of cell survival compared to untreated cells. Regarding Random copolymers, their toxicity shows a tendency to increase with their size (IC_50_ values from 124.7 to >400 µg/mL) (Figure 2a and Table 3). 

With Diblock copolymers, the relationship between their size and their toxicity is less evident, with IC_50_ values ranging from 161.8 to 250.6 µg/mL (Figure 2b and Table 3). 

In addition to IC_50_ values, the innocuity of the copolymers must be evaluated through the determination of their therapeutic index (TI), i.e., the relation between their IC_50_ on BEAS-2B and their MIC on the bacterial strains. It is admitted that the TI must be higher than 1 and that the higher the TI, the safer the compound is. As shown in Table 3, for all tested copolymers and regardless of the bacterial strain, all TI are >1. The highest TI is >64 for the Random copolymers (Random_2800_ on *M. abscessus* S; Table 3) and 20 for the Diblock copolymers (Diblock_9500_ on *S. aureus*; Table 3).

Based on the MIC and the TI values, Random_10200_, Random_15000_, Random_23900_, and Diblock_9500_ copolymers, which display good antibacterial activity against the three bacteria associated with low toxicity (Figure 3), were selected for further investigations. It has to be noted that, although Random_2800_ had the lowest cytotoxicity on BEAS-2B cells (i.e., IC_50_ > 400 µg/mL) and a very potent activity on *M. abscessus* S (MIC of 6.25 µg/mL), this copolymer was found to be weakly active on *P. aeruginosa* and *S. aureus*, explaining why it was not further evaluated in the present study.

### 2.3. Determination of Time-Kill Kinetics

A time-kill assay was performed to evaluate the kinetics of action of the four selected copolymers over time (Figure 4). Bacteria were incubated with each copolymer at 4× MIC and took off at different times to be spread on agar plates for colonies forming units (CFU) counting. In addition to the copolymers, the quaternary ammonium cetyltrimethylammonium bromide (CTAB) was used as a control, this cationic surfactant being able to rapidly insert into the bacterial membrane, leading to a fast membranolytic effect [49,50,51]. Conventional bactericidal antibiotics were also used for comparison (i.e., clarithromycin for *M. abscessus*, imipenem for *P. aeruginosa*, and mupirocin for *S. aureus*). Results show that the copolymers Random_15000_ and Random_23900_ display a strong and rapid killing effect against the three bacteria tested. Indeed, complete killing of all bacteria (6 log reduction) is achieved in 120 min, with already a ≥3 log reduction in the CFU count within 60 min (Figure 4). This fast killing activity is similar to the one of CTAB, reinforcing the idea that these copolymers, as most AMPs, act through a membranolytic effect. Importantly, these copolymers act more rapidly and strongly than control conventional bactericidal antibiotics leading to less than 1 log reduction in CFU count within 60 min and reaching approximatively 2–4 log reduction after 240 min exposure (Figure 4). The Random_10200_ compound also exhibits a fast bactericidal activity on *P. aeruginosa* and *S. aureus* with a >5 log CFU/mL reduction within 120 min (Figure 4b and c, respectively) but with no killing effect on *M. abscessus* S during the time course of the assay (Figure 4a). Concerning Diblock_9500_, although a quick bactericidal activity is observed against *S. aureus* with a >5 log CFU/mL reduction in 120 min (Figure 4c), the results show that this copolymer possesses no bactericidal activity during the assay against *M. abscessus* S (Figure 4a) and only a slow killing activity on *P. aeruginosa*, which is similar to the conventional bactericidal antibiotic imipenem (Figure 4b).

### 2.4. Mechanism of Action

Since the killing assay suggested that most of the tested copolymers display a rapid bactericidal activity similarly to the membranolytic detergent CTAB, the effect of copolymer on membrane integrity was then evaluated using a propidium iodide assay (Figure 5). Indeed, the cell impermeable dye propidium iodide can enter bacteria and form a highly fluorescent complex with DNA/RNA only if the cell membrane is damaged, allowing for monitoring of the membrane’s integrity. The membranolytic detergent CTAB was used as a positive control giving 100% permeabilization [49,50,51,52]. In *M. abscessus* S, the copolymers induce a very rapid permeabilization, with more than 50% permeabilization in 5 min (Figure 5a). Within 120 min, more than 90% permeabilization are obtained with Random_15000_ and Random_23900_, the permeabilization activity being lower for Random_10200_ and Diblock_9500_ (approximately 34% and 62% permeabilization, respectively) (Figure 5a). In *P. aeruginosa*, the permeabilization activity of Random_15000_, Random_23900_, and Random_10200_ is more progressive, but these three compounds induce more than 80% permeabilization within 120 min exposure (Figure 5b). On the contrary, Diblock_9500_, which does not have any bactericidal activity on *P. aeruginosa* (Figure 5b), is unable to permeabilize *P. aeruginosa* cells during the 120 min of the assay (Figure 5b). Last, on *S. aureus*, the four copolymers induce a relatively rapid permeabilization, with more than 65% permeabilization within 30 min and ≥80% after 120 min exposure (Figure 5c).

### 2.5. Insertion in Lipid Monolayers

The ability of the selected copolymers to insert into the bacterial membranes was further evaluated using Langmuir lipid monolayer assay (Figure 6). Total membrane lipids were first extracted from each strain. The extracted total lipids were spread at the air-water interface to reach a surface pressure of 30 ± 0.5 mN/m corresponding to a lipid packing density theoretically equivalent to that of the outer leaflet of the bacterial cell membrane [53]. Increasing concentrations of copolymer were then injected into the aqueous phase below the lipid monolayers. Their ability to insert into the films, causing variations in the surface pressure (deltaPi), was measured using surface microtensiometer. In accordance with propidium iodide assay results (Figure 6), data show that Random_10200_, Random_15000_, and Random_23900_ are able to efficiently insert into lipids extracted from *M. abscessus* S, *P. aeruginosa*, and *S. aureus*, with a maximal insertion observed at concentrations close or superior to the MIC values, i.e., at 1 to 4× MIC (Figure 6). The same result is obtained in the case of Diblock_9500_, except for the total lipids extracted from *P. aeruginosa* for which lipid insertion is weaker, reflecting the fact that Diblock_9500_ only causes a limited membrane permeabilization in *P. aeruginosa* (Figure 6b).

### 2.6. Resistance Induction Assay

Results obtained above showed that the selected copolymers target the bacterial membrane. Such membranolytic activity, also present in cationic AMPs, suggests that these copolymers should be less prompt at selecting resistant bacteria compared to conventional antibiotics targeting enzyme/proteins for which mutations may induce resistance. To confirm this hypothesis, the appearance of resistant bacteria was evaluated through repeated MIC testing during 30–45 days (Figure 7).

Results show that none of the tested copolymers are able to cause increased MIC values typical of an induction of resistance even after 30–45 days of continuous exposure. By contrast, conventional antibiotics known to be prompt to cause resistance in bacteria select resistant strains after 10–15 days of exposure. Importantly, copolymers fully retain their activity against those generated antibiotic-resistant bacteria with unchanged MIC values.

### 2.7. Anti-Biofilm Activity

The difficulty to treat *P. aeruginosa* infections is essentially linked to its ability to form biofilms [54]. As a consequence, the formation of highly structured biofilms allows *P. aeruginosa* to compete, survive, and overtakes other bacteria in the lung of CF patients colonized by a great variety of microorganisms [55]. Therefore, the effect of selected copolymers on biofilm formation was measured. As shown in Figure 8, Random_10200_, Random_15000_, Random_23900_, and Diblock_9500_ dose-dependently prevent biofilm formation. Diblock_9500_ is the most interesting one with a statistically significant preventing effect observed with concentration as low as 0.25× MIC. Unfortunately, copolymers were found inactive against preformed/established biofilms.

## 3. Discussion

Currently, a major global challenge for health is to fight against antibiotic resistance. Indeed, we are facing the dramatic appearance in resistance to one or more antibiotics in a vast number of bacteria infecting humans, combined with the decline in the discovery of new antibiotics [56,57,58]. Furthermore, patients infected with antibiotic-resistant bacteria have generally worse clinical outcomes, higher probability of death, and consume more healthcare resources than patients infected with non-resistant bacteria [56]. In other words, antibiotic resistance increases morbidity, mortality, and hospital costs [56,57,58]. Therefore, there is an urgent need to develop new antibiotics, notably antibiotics against which bacteria develop low or no resistance [56,57,58]. This problem of drug resistance is particularly important for CF patients for which lung complications are often the cause of death, notably lung colonization by pathogenic bacteria [4]. In this study, we identified four methacrylate-based cationic copolymers that could have a great potential for the treatment of CF patients. These copolymers possess a methacrylate backbone, with DMAEMA as the hydrophilic monomer and BMA the hydrophobic monomer. Three of the selected copolymers are Random copolymers (Random_10200_, Random_15000_, and Random_23900_) while one is a Diblock copolymer (Diblock_9500_). All four copolymers display low cytotoxicity on human lung epithelial cells and exhibit a very good antibacterial activity with relatively high TI values on *P. aeruginosa* and *S. aureus*, the most frequently found bacteria in CF patients. The copolymers are also active against *M. abscessus* S variant, a bacteria often found later in the disease course and nicknamed the “antibiotic nightmare” [6,7,17]. Among the selected copolymers, Random_15000_ and Random_23900_ have a strong and rapid killing effect on *P. aeruginosa*, *S. aureus* as well as on *M. abscessus* S. Our data strongly support that this bactericidal activity could be due to the permeabilization of the bacterial membrane leading to their lysis and death. This certainly results from the interaction between the cationic part of the copolymer (due to the presence of DMAEMA) and the negatively charged surface of the bacteria, before insertion of the copolymer into the bacterial membrane thanks to its hydrophobic part (due to the presence of BMA). The fact that these copolymers have no effect on *M. abscessus* R variant, suggests that the surface-exposed GPL present only in the S variant may play a key role in the interactions governing the antibacterial activity of the copolymers. It is now admitted that CF patients are first infected by *M. abscessus* S variant, which could evolve into the R form able to form cords and leading to a severe and almost incurable pulmonary infection. In this context, these copolymers could represent a promising alternative strategy to kill this mycobacterium at the early stage of the infection, thus preventing the formation of cords by R variants. Moreover, given their pore-forming mode of action, the probability for the bacteria to develop resistance against these compounds is very low, rendering these copolymers particularly interesting in the treatment of CF patients. This was confirmed by the absence of appearance of resistance against these four copolymers after 30–45 days continuous exposure as previously described for similar copolymers [47]. In addition, the copolymers were found active on bacteria trained to become resistant to conventional antibiotics. Interestingly, it has to be noted that Diblock_9500_, although found membranolytic for *M. abscessus* S and *S. aureus*, does not cause membrane permeabilization in *P. aeruginosa*. This suggests another mechanism of action for this copolymer on this strain that would be investigated in the future. Finally, similar to numerous AMPs [56,59,60], the tested copolymers were able to prevent biofilm formation implying a major impact of our molecule at the early stage of the infection. Unfortunately, selected copolymers were inactive on established biofilms, a limitation also observed with several AMPs [56,59,60], only few AMPs being shown to both inhibit biofilm formation and disrupt established ones [59].

Overall, this study shows that the Random and Diblock copolymers display the same strengths than AMPs in the treatment of CF-associated bacteria [60], notably their capacity to rapidly kill bacteria through a membranolytic activity, without causing resistance. However, unlike AMPs, Random and Diblock copolymers are insensitive to proteases, not immunogenic, and easy to produce in a large industrial scale. Future studies, including *in vivo* assays, will be necessary however to confirm their potential in the treatment of bacterial infections in CF patients.

## 4. Materials and Methods

### 4.1. Synthesis and Characterization of the Cationic Amphiphilic Polymers

The synthesis of the different Random and Diblock copolymers by nitroxide-mediated polymerization (NMP) has already been described [47]. A brief description of their synthesis is given below. The synthesis of the PBMA-*b*-PDMAEMA Diblock copolymers is a two-step procedure. BMA, acrylonitrile (ACN, comonomer of the NMP, 10mol% vs. BMA), BlocBuilder (initiator) and SG1 (control agent) were introduced in a 100 mL two-necked round bottom glass flask fitted with a magnetic stir bar. After 30 min of nitrogen bubbling at RT, the flask was heated at 90 °C under Argon. The reaction was stopped when the conversion is close to 45%. The conversion was determined by ^1^H NMR spectroscopy. The fist block was recovered by precipitation into MeOH/H_2_O and analyzed by SEC/DMF to obtain the molecular weight (*M*_n_) and dispersity (Ð) values. After drying, the PBMA was used as a macroinitiator for the polymerization of the DMAEMA to produce the corresponding PBMA-*b*-PDMAEMA Diblock copolymer. Typically, PBMA, DMAEMA, ACN (10 mol% vs. DMAEMA), 1,4-dioxane and SG1 were introduced in a 25 mL two-necked round bottom glass flask fitted with a magnetic stir bar. The solution was deoxygenated by nitrogen bubbling for 30 min at room temperature before heating at 90 °C under argon. The reaction was stopped when the conversion reached 40%. The final PBMA-*b*-PDMAEMA Diblock copolymer was purified by precipitation in pentane and dried under vacuum before analysis by ^1^H NMR spectroscopy and SEC/DMF to obtain the composition (with *F*_DMAEMA_ the DMAEMA molar composition in the final copolymer) and the number-average molecular weight *M*_n_), respectively. The synthesis of the Random copolymers is performed in one step. Typically, BMA, ACN, DMAEMA, BlocBuilder, and SG1 were introduced in a 25 mL two-necked round-bottom flask fitted with a magnetic stir bar. After 30 min of deoxygenation by argon bubbling at room temperature, the solution was heated at 90 °C under argon. Samples were taken periodically to determine the conversion (^1^H NMR spectroscopy/CDCl_3_/400 MHz) and the molecular weights (SEC/DMF). The reaction was stopped when the conversion was close to 50%. The final P(BMA-*co*-DMAEMA) Random copolymer was isolated by precipitation in pentane, then dried under vacuum before analysis. The permanent quaternization of the Random and Diblock copolymers (quaternary ammonium ion) was achieved by methylation of the amine group using methyl iodide (MeI). Typically, after dissolution of the copolymers in THF, 2 mol equivalents of MeI *versus* the number of amine function of DMAEMA were slowly added. After 18 h of stirring at room temperature, the antibacterial copolymers were recovered by precipitation in diethyl ether and then dried under vacuum.

### 4.2. Bacterial Strains and Growth Conditions

Reference strains used in this study were obtained either from the American Type Culture Collection (ATCC, Molsheim Cedex France) or the French Pasteur Institute (CIP, Paris, France). *Pseudomonas aeruginosa* (ATCC 9027) and *Staphylococcus aureus* (ATCC 6538P) were routinely grown on Luria Bertani (LB) agar plates and LB broth at 37 °C. *M. abscessus* S and R (CIP 104536^T^) were grown in 7H9 medium supplemented with 0.05% Tween 80, 0.2% glycerol, and 10% oleic acid–albumin–dextrose–catalase (OADC enrichment; BD Difco) (7H9-S^OADC^) at 37 °C under agitation (200 rpm).

### 4.3. Minimal Inhibitory Concentration (MIC) Determination

Antimicrobial activity of the polymers was evaluated by determination of their MIC, i.e., the lowest concentration of polymer that inhibits visible growth of the organism compared to a control, following the National Committee of Clinical Laboratory Standards (NCCLS, 1997). Briefly, for *P. aeruginosa* and *S. aureus*, single colonies of the different bacterial strains cultured on agar plates were used to inoculate 3 mL of Mueller–Hinton broth (MH). Tubes were incubated overnight (for approximately 16 h) at 37 °C under stirring (200 rpm). The bacterial suspensions were adjusted to OD_600 nm_ = 1 with the medium, then diluted 1/100 in 3 mL of fresh medium and incubated at 37 °C, 200 rpm until bacteria reached log phase growth (OD_600 nm_ approximately 0.6). Then, 100 μL of bacterial suspension (10^5^ cells/mL) were added per well into sterile polypropylene 96 wells microplates (Greiner BioOne, Dominique Dutscher, Brumath, France) containing a serial two-fold dilution of polymers’ increasing concentrations ranging from 0 to 400 µg/mL. Plates were incubated at 37 °C for 18–24 h and OD_600 nm_ measurements were taken using microplate reader (Biotek, Synergy Mx, Colmar, France) to determine the MIC. Experiments were conducted in independent triplicate (n = 3). Susceptibility testing for *M. abscessus* was performed using the Middlebrook 7H9 broth microdilution method in 96-well flat-bottom Nunclon Delta Surface microplates with lid (Thermo-Fisher Scientific, Illkirch-Graffenstaden, France). Briefly, log-phase bacteria were diluted to a cell density of 5 × 10^6^ cells/mL in 7H9-S^OADC^ medium; then, 100 μL of this bacterial suspension was added to each well containing 100 μL of serial two-fold dilutions of the copolymers or controls to a final volume of 200 μL. Plates were incubated at 37 °C for 3–5 days, and MIC was recorded by absorbance measurement at 560 nm. At least three independent experiments were conducted for each copolymer.

### 4.4. Evaluation of the Cytotoxic Effect of Polymers Using Human Cells

The toxicity of the polymers on human cells was evaluated as previously described [61] using human non tumorigenic bronchial epithelial cells BEAS-2B (ATCC CRL-9609) as model. Cells were routinely grown on 25 cm^2^ flasks in DMEM medium supplemented with 10% Fetal Bovine Serum (FBS) and 1% antibiotics (all from Thermo Fisher Scientific, Illkirch-Graffenstaden, France) in a 5% CO_2_ incubator at 37 °C. For cytotoxicity assay, cells were detached using trypsin–EDTA solution (from Thermo Fisher Scientific, Illkirch-Graffenstaden, France). Cells were counted using Malassez counting chamber and seeded into 96-well cell culture plates (Greiner bio-One, Dominique Dutscher, Brumath, France) at approximately 10,000 cells per well. The cells were left to grow for 48–72 h at 37 °C in a 5% CO_2_ incubator until they reached confluence. Medium from wells was then aspirated and cells were treated with 100 μL of culture medium containing increasing concentrations of polymers obtained by serial dilution (from 0 to 400 µg/mL, 1:2 dilution). After 48 h incubation at 37 °C in a 5% CO_2_ incubator, the cell viability was evaluated using a resazurin based in vitro toxicity assay kit (Sigma-Aldrich, Lyon, France), following manufacturer’s instructions. Briefly, the culture medium was removed and cells were treated with 100 μL of the resazurin diluted 1:10 in sterile phosphate buffer saline (PBS) containing calcium and magnesium (PBS^++^, pH 7.4). After 4 h incubation at 37 °C, fluorescence intensity (λ_ex_/λ_em_ = 530/590 nm) was measured using a microplate reader (Biotek, Synergy Mx, Colmar, France). The fluorescence values were normalized by the negative controls and expressed as percentage of viability. The IC_50_ value of the copolymers (i.e., the concentration of polymers causing a reduction of 50% of the cell viability as compared to the control) was calculated using GraphPad^®^ Prism 7 software (San Diego, CA, USA). Experiments were conducted in triplicate (n = 3).

### 4.5. Time–Kill Assays 

Time-kill assay was performed as previously described [60,62]. Log-phase bacteria were incubated in wells of a 96 wells microplate at a final concentration of 10^6^ cell/mL with either medium alone (growth control), or 150 µM cetyltrimethylammonium bromide (CTAB), or copolymer at 4× MIC. Conventional bacteriolytic antibiotics (at 4× MIC) were used as controls, i.e., clarithromycin for *M. abscessus*, imipenem for *P. aeruginosa*, and mupirocin for *S. aureus*. The plate was incubated at 37 °C in a humidity chamber. At defined time intervals (5, 15, 30, 60, 120 and 240 min), samples were taken from each well and serial dilutions were prepared and plated on agar plates. The number of CFU/mL was then evaluated after 5 days (for *M. abscessus)* or overnight (for *P. aeruginosa* and *S. aureus*) incubation at 37 °C. Experiments were conducted at least in triplicate (n ≥ 3).

### 4.6. Bacterial Membrane Permeabilization Assay

Membrane permeabilization by copolymers was evaluated using propidium iodide, a cell-impermeable DNA/RNA probe, as previously described [63]. Logarithmic growing bacterial suspensions of *P. aeruginosa* or *S. aureus* were prepared from overnight bacterial suspension by 1 in 100 dilutions. After 3 h incubation at 37 °C, 200 rpm, bacterial suspensions were centrifuged for 5 min at 3000 rpm. Cell pellets were then resuspended in MH at approximately 10^9^ cells/mL. Similar protocol was used for *M. abscessus* with log-phase bacteria, the cell pellet being resuspended in 7H9-S^OADC^. Propidium iodide (Sigma Aldrich, Lyon, France) was added to these suspensions at a final concentration of 1 mg/mL. Then, 100 μL of this suspension were transferred into Greiner polypropylene 96-well black plates (Greiner bio-One, Dominique Dutscher, Brumath, France) containing copolymers at 4× MIC, or 150 µM CTAB as positive control leading to 100% permeabilization. A negative control was included, with bacteria and propidium iodide but without copolymer. Bacteria were incubated at 37 °C and fluorescence (λ_ex_/λ_em_ = 530/590 nm) was recorded over time using a microplate reader (Tecan Spark™, Tecan Group Ltd., France for *M. abscessus*, and Biotek, Synergy Mx, Colmar, France for *P. aeruginosa* and *S. aureus*). Results were expressed in percentage of permeabilization compared to the maximum fluorescence obtained with CTAB, after subtraction of fluorescence of the negative control. All experiments were independently performed at least three times.

### 4.7. Induction of Resistance Assay

Induction of bacterial resistance by copolymers was evaluated as previously described [60,62,64]. Briefly, bacterial cultures were continuously exposed to copolymers or conventional antibiotics over 45 consecutive days for *M. abscessus* and 30 consecutive days for *P. aeruginosa* and *S. aureus*, through repeated broth microdilution susceptibility testing and determination of MIC values. After each determination of the MIC values, the well containing the highest concentration of copolymer allowing a bacterial growth was diluted 1:1000 in medium and used as inoculum for the next MIC assay. 

### 4.8. Lipid Insertion Assay

Interaction of polymers with bacterial lipids was measured using reconstituted lipid monolayer at the air-water interface as previously described [65]. Total lipids were extracted from bacterial suspensions of *P. aeruginosa* or *S. aureus* using Folch procedure [61,64]. With *M. abscessus*, total lipids were extracted as previously described [66,67] with slight modifications. Briefly, total lipids from dry extract were incubated for 16 h with CHCl_3_-MeOH (1:2; *v/v*) at room temperature under shaking. Residual lipids were re-extracted, first with CHCl_3_-MeOH (1:1, *v/v*) and then for 16 h with the same solvents but using an alternative ratio (2:1, *v/v*). All the three organic phases were pooled and concentrated under reduced pressure. Samples were re-suspended in CHCl_3_-MeOH (3:1, *v/v*), and washed with 0.3% (*w/v*) NaCl solution. Organic and aqueous phases were separated by centrifugation 5000× *g* for 15 min and the organic phase was dried over MgSO4. Each sample of extracted total lipids was dried, resolubilized in CHCl_3_-MeOH (2:1, *v/v*) and stored at –20 °C under nitrogen. For lipid interaction assay, total lipid extract was spread using a 50 µL Hamilton’s syringe at the surface of 800 µL of sterile ultra-pure water creating a lipid monolayer at the air-water interface. Lipids were added until the surface pressure reached 30 ± 0.5 mN/m, a value corresponding to a lipid packing density theoretically equivalent to that of the outer leaflet of the cell membrane [53]. After a delay of 5–10 min to allow for solvent evaporation and stabilization of the initial surface pressure, 8 µL of polymers more or less diluted in sterile ultra-pure water were injected into the ultra-pure water sub-phase (volume 800 µL) using a 10 µL Hamilton syringe to reach the desired final concentration of polymer. The variation of the surface pressure caused by polymer insertion was then continuously monitored using a fully automated microtensiometer (µTROUGH SX, Kibron Inc., Helsinki, Finland) until reaching equilibrium (within 90 min of continuous recording). All experiments were carried out in a controlled atmosphere at 20 °C ± 1 °C, and data were analyzed using the Filmware 2.5 program (Kibron Inc., Helsinki, Finland).

### 4.9. Antibiofilm Formation Assay

*P. aeruginosa* ATCC 9027 was grown in Mueller Hinton (MH) broth at 37 °C with agitation at 200 rpm until OD_600nm_ reached 0.2–0.3. Cells were then diluted to 10^5^ CFU/mL in MH and 150 µL of cell suspension was added per well of the Calgary biofilm device (CBD). Sterile polymers were added at a maximum concentration of 400 µg/mL and twofold series dilutions were performed in cell suspension to a minimal concentration of 0.8 µg/mL. As it has been demonstrated that an edge effect affects the biofilm formation, all the edges of the plates were filled with MH and used as sterility and negative controls [59]. Conversely, untreated wells containing bacterial suspension were used as a positive control for biofilm formation. After 24 h of incubation at 37 °C with agitation reduced to 110 rpm, the lid of the CBD was removed and the pegs were rinsed twice in PBS and then fixed in 100% methanol for 15 min. Pegs were rinsed with PBS once more and then air-dried before being stained with crystal violet at 0.2% for 15 min. Pegs were rinsed with PBS twice before being air-dried and then destained in methanol for 15 min before being discarded. Absorbance at 570 nm was measured to evaluate biofilm formation. All the rinsing, staining, and destaining steps were performed in 96-well plates with a volume of 200 µL per well in order for the biofilm to be totally soaked. All experiments were performed in independent triplicates.

### 4.10. Antibiofilm Disruption Assay

*P. aeruginosa* ATCC 9027 was grown in Luria Bertani (LB) broth at 37 °C with agitation at 200 rpm until OD_600nm_ reached 0.2–0.3. Cells were then diluted to 10^5^ CFU/mL in LB, and 150 µL of cell suspension was added per well of the Calgary biofilm device without the addition of polymers and taking into account the edge effect previously mentioned. After 24 h of growth at 37 °C under stirring at 110 rpm, the lid of the CBD was transferred into a new 96-well plate containing fresh MH media supplemented either with polymers at concentrations ranging from 0.8 to 400 µg/mL or not. Wells were filled with 200 µL to make sure the pegs were totally soaked. Then, after 24 h of incubation in the same condition, the pegs were rinsed, fixed, stained, and destained as described above. A volume of 300 µL was used per well. Absorbance at 570 nm was measured to evaluate biofilm formation and disruption.

### 4.11. Statistical Analysis

All assays were performed in triplicate (n = 3). Data were plotted and analyzed using GraphPad^®^ Prism 7 software (San Diego, CA, USA). Statistical significancy of the data was evaluated using a *Mann-Whitney test.*

## 5. Conclusions

In this study, various Random and Diblock methacrylate-based cationic copolymers ranging from 2800 to 23,900 g/mol were tested against the three major bacterial species colonizing CF patients (i.e., *P. aeruginosa*, *S. aureus*, and *M. abscessus*). Evaluation of the antibacterial activity and innocuity on human cells allowed the identification of four copolymers, i.e., Random_10200_, Random_15000_, Random_23900_, and Diblock_9500_, as the best molecules. Mechanistic approaches demonstrated that the copolymers, due to their amphipathic character and cationic charge, preferentially interact with the bacterial membrane, leading to a rapid membranolytic effect and the death of the bacteria. Random and Diblock methacrylate-based cationic copolymers thus display the same strengths as AMPs against bacteria, notably their fast killing action related to a membranolytic activity preventing the development of resistance and allowing them to remain active on bacteria already resistant to antibiotics. In addition, their insensitivity to proteases, the absence of immunogenicity, and their ease of production make those copolymers good candidates as antibacterial agents. Although in vivo confirmation will be necessary, our results already demonstrate the potential value of the described antimicrobial amphiphilic cationic copolymers in the fight against bacteria, particularly the ones infecting CF patients.

## Figures and Tables

**Figure 1 antibiotics-12-00120-f001:**
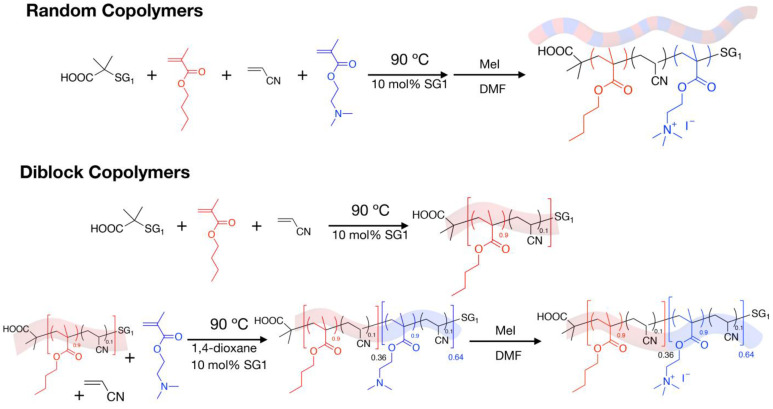
Synthetic routes and structures of Random and Diblock methacrylate-based copolymers. The synthesis of the copolymers detailed in the Materials and Methods section involves one (for Random) or two steps (for Diblock) synthesis. In all cases, the synthesis involves two monomers: butyl methacrylate (BMA, hydrophobic monomer, in red) and dimethylaminoethyl methacrylate (DMAEMA, hydrophilic monomer, in blue), and acrylonitrile (ACN), BlocBuilder (initiator), and SG1 (control agent) to initiate the polymerization. The permanent quaternization of the Random and Diblock copolymers (quaternary ammonium ion) is achieved by the methylation of the amine group in the presence of methyl iodide (MeI).

**Figure 2 antibiotics-12-00120-f002:**
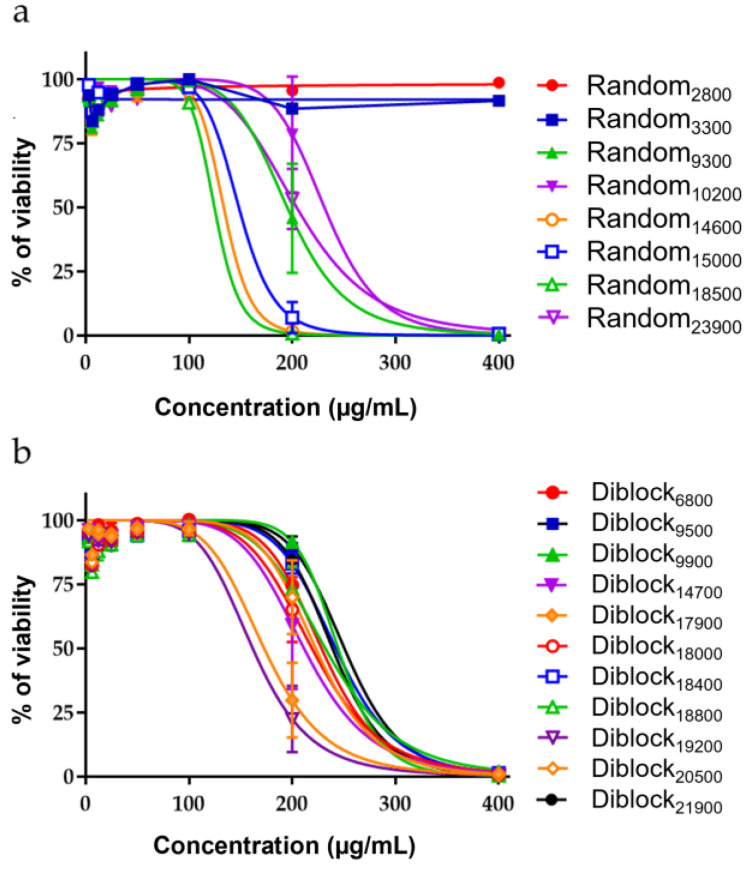
Evaluation of the toxicity of Random and Diblock copolymers. Non tumorigenic human airway epithelial cells (BEAS-2B) were exposed for 48 h to increasing concentrations of Random (**a**) or Diblock (**b**) copolymers. At the end of the incubation, the cell viability (expressed as % of viability of control cells) was measured using the resazurin assay. Data were analyzed using GraphPad Prism (n = 3).

**Figure 3 antibiotics-12-00120-f003:**
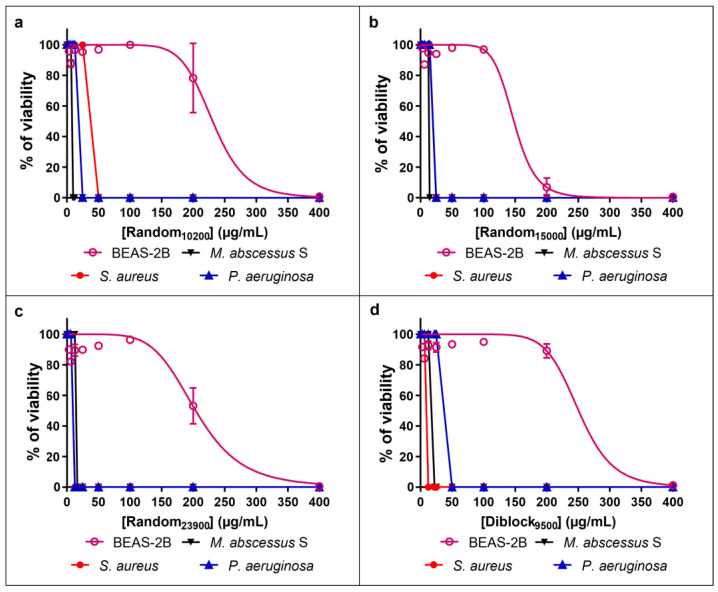
Comparative toxicity and antibacterial activity of the selected copolymers. The four more interesting copolymers, i.e., Random_10200_ (**a**), Random_15000_ (**b**), Random_23900_ (**c**), and Diblock_9500_ (**d**), were tested in terms of toxicity on human cells (BEAS-2B cells) and activity on bacteria. Results are expressed as the mean ± SD of the percentage of cell viability (n = 3).

**Figure 4 antibiotics-12-00120-f004:**
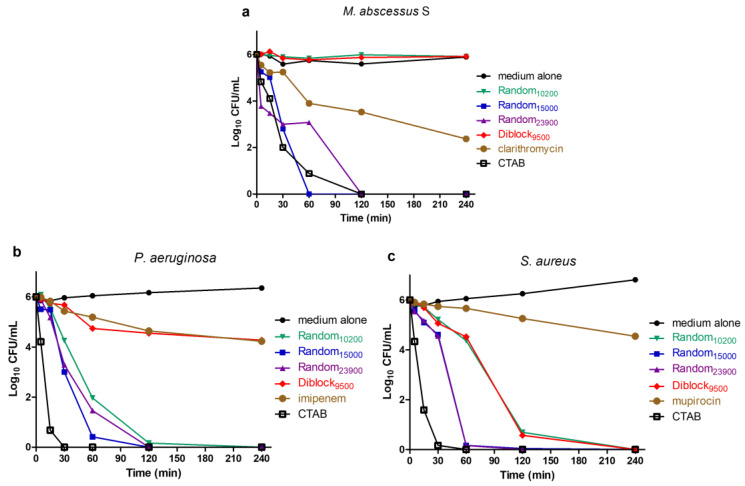
Time-dependent killing activity of the copolymers. Bacterial strains (*M. abscessus* S (**a**), *P. aeruginosa* (**b**) and *S. aureus* (**c**), at approximatively 10^6^ CFU/mL initial bacterial density) were incubated with Random or Diblock copolymers at 4× MIC concentrations. Results are expressed as the mean of logarithm of the number of CFU/mL as a function of time (n = 3). To facilitate visualization, SD are not represented. Conventional bactericidal antibiotics (i.e., clarithromycin, imipenem, and mupirocin for *M. abscessus* S, *P. aeruginosa*, and *S. aureus*, respectively) in addition to CTAB (a quaternary ammonium inducing bacterial killing by membranolytic effect) were used as controls.

**Figure 5 antibiotics-12-00120-f005:**
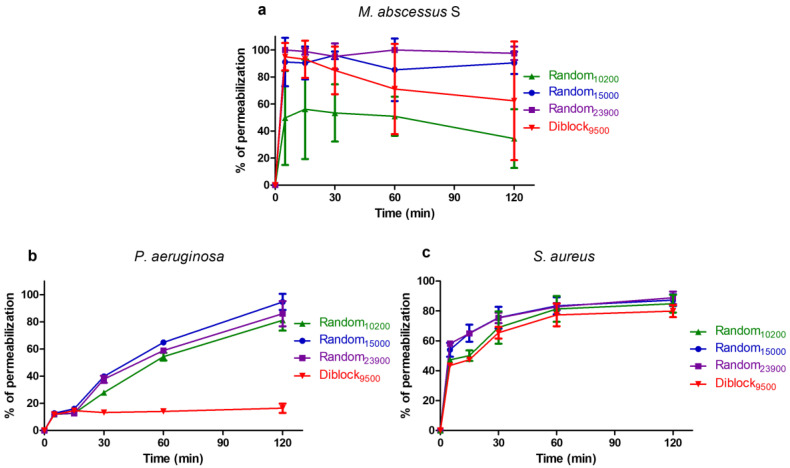
Effect of the copolymers on bacterial membrane integrity. The effect of the copolymers on membrane integrity of *M. abscessus* S (**a**), *P. aeruginosa* (**b**), and *S. aureus* (**c**) was measured as described in the Materials and Methods. Results are expressed as mean percentage of permeabilization ± SD, the maximum fluorescence obtained with CTAB (at 150 µM) being used as a positive control giving 100% of permeabilization (n = 3).

**Figure 6 antibiotics-12-00120-f006:**
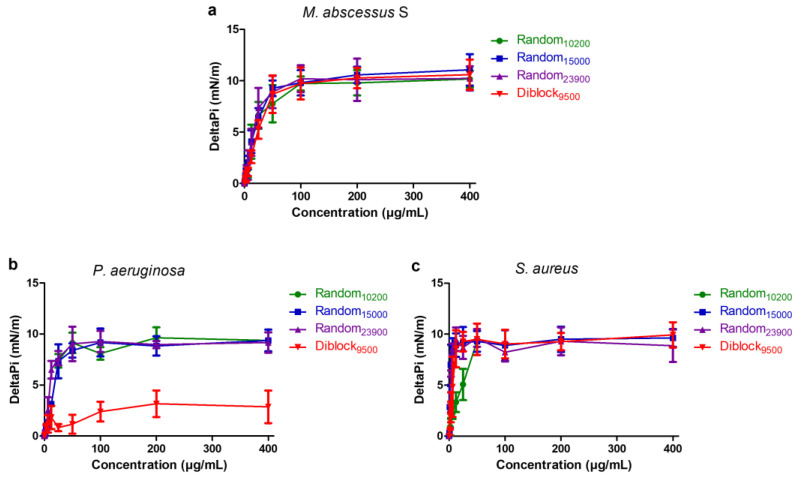
Evaluation of the insertion of the copolymers in lipid monolayers. The insertion of the copolymers in lipid monolayers from *M. abscessus* S (**a**), *P. aeruginosa* (**b**), and *S. aureus* (**c**) was evaluated by measuring the surface pressure (deltaPi) as a function of copolymer concentration. Mean ± SD of deltaPi (mN/m) is represented (n = 3).

**Figure 7 antibiotics-12-00120-f007:**
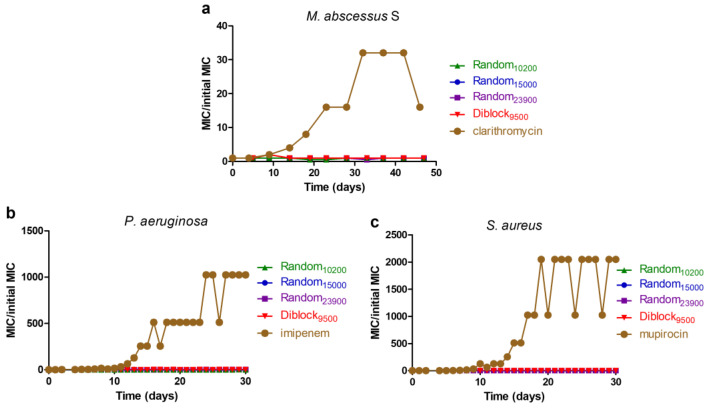
Evaluation of the acquisition of resistance caused by copolymers. Induction of resistance caused by copolymers was evaluated by repeated MIC testing over 30 or 45 days exposure of *M. abscessus* S (**a**), *P. aeruginosa* (**b**), and *S. aureus* (**c**). Conventional antibiotics prompt to cause resistance were used as positive controls (i.e., clarithromycin, imipenem, and mupirocin for *M. abscessus* S, *P. aeruginosa*, and *S. aureus*, respectively).

**Figure 8 antibiotics-12-00120-f008:**
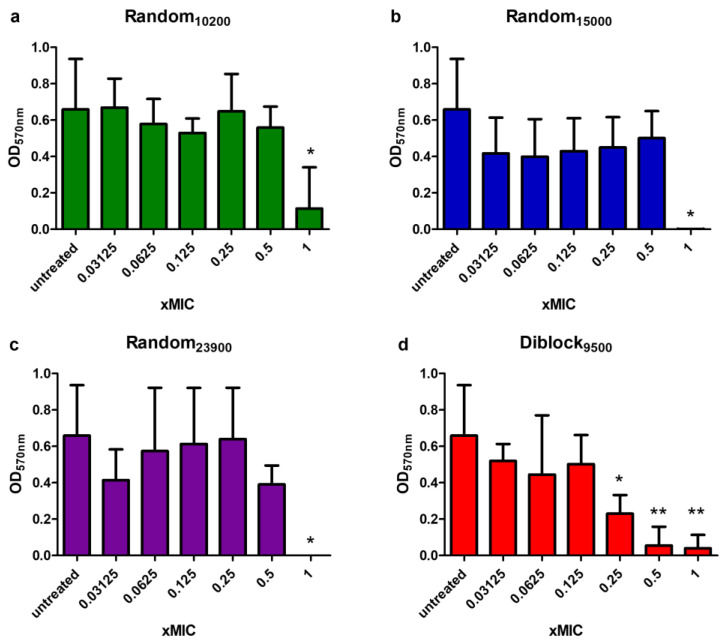
Quantification of anti-biofilm activity of copolymers. OD values at 570 nm from which blank values have been removed, were measured after peptide treatment at different concentrations (expressed as a function of MIC) before biofilm formation. Mean ± SD (n ≥ 3) is represented for Random_10200_ (**a**), Random_15000_ (**b**), Random_23900_ (**c**), and Diblock_9500_ (**d**). *: *p* < 0.05 and **: *p* < 0.01 compared to untreated cells (Mann–Whitney test).

**Table 1 antibiotics-12-00120-t001:** Characteristics of the Random and Diblock copolymers with *M*_n_ the number average molecular weight and *F*_DMAEMA_ the DMAEMA molar ratio in the final copolymer.

Copolymer Name	*M*_n_ (g/mol) (SEC/DMF)	*F* _DMAEMA_
Random_2800_	2800	0.67
Random_3300_	3300	0.63
Random_9300_	9300	0.65
Random_10200_	10,200	0.64
Random_14600_	14,600	0.65
Random_15000_	15,000	0.56
Random_18500_	18,500	0.68
Random_23900_	23,900	0.61
Diblock_6800_	6800	0.65
Diblock_9500_	9500	0.62
Diblock_9900_	9900	0.72
Diblock_14700_	14,700	0.78
Diblock_17900_	17,900	0.62
Diblock_18000_	18,000	0.8
Diblock_18400_	18,400	0.64
Diblock_18800_	18,800	0.57
Diblock_19200_	19,200	0.74
Diblock_20500_	20,500	0.64
Diblock_21900_	21,900	0.58

**Table 2 antibiotics-12-00120-t002:** Evaluation of the antimicrobial activity of the Random and Diblock copolymers. MIC of Random and Diblock copolymers were measured as described in Materials and Methods section and are expressed in µg/mL (n ≥ 3).

Copolymer Name	*M. abscessus* S	*P. aeruginosa*	*S. aureus*
Random_2800_	6.25	200	200
Random_3300_	25	400	200
Random_9300_	25	25	50
Random_10200_	12.5	25	50
Random_14600_	25	25	12.5
Random_15000_	25	25	25
Random_18500_	25	12.5	12.5
Random_23900_	25	12.5	12.5
Diblock_6800_	25	100	12.5
Diblock_9500_	50	50	12.5
Diblock_9900_	50	50	12.5
Diblock_14700_	50	50	12.5
Diblock_17900_	100	50	12.5
Diblock_18000_	50	50	12.5
Diblock_18400_	100	50	25
Diblock_18800_	100	50	12.5
Diblock_19200_	50	50	12.5
Diblock_20500_	50	100	12.5
Diblock_21900_	100	50	12.5

**Table 3 antibiotics-12-00120-t003:** Evaluation of the innocuity of Random and Diblock copolymers. IC_50_ values of Random and Diblock copolymers (expressed in µg/mL) on BEAS-2B viability were calculated using data from Figure 2a. Therapeutic Index (TI) was calculated by dividing the IC_50_ of the copolymer by their MIC value for each bacterial strain.

Copolymer Name	IC_50_ BEAS-2B	TI *M. abscessus* S	TI *P. aeruginosa*	TI *S. aureus*
Random_2800_	>400	>64	>2	>2
Random_3300_	>400	>16	>1	>2
Random_9300_	195.1	7.8	7.8	3.9
Random_10200_	231.1	18.5	9.2	4.6
Random_14600_	133.7	5.3	5.3	10.7
Random_15000_	148.3	5.9	5.9	5.9
Random_18500_	124.7	5.0	10.0	10.0
Random_23900_	204.2	8.2	16.3	16.3
Diblock_6800_	228.4	9.1	2.3	18.3
Diblock_9500_	250.6	5.0	5.0	20.0
Diblock_9900_	244.5	4.9	4.9	19.6
Diblock_14700_	211.5	4.2	4.2	16.9
Diblock_17900_	172.8	1.7	3.4	13.8
Diblock_18000_	219.2	4.4	4.4	17.5
Diblock_18400_	242.1	2.4	4.8	9.7
Diblock_18800_	232.1	2.3	4.6	18.6
Diblock_19200_	161.8	3.2	3.2	12.9
Diblock_20500_	222.7	4.4	2.2	17.8
Diblock_21900_	239	2.4	4.8	19.1

## Data Availability

Data available on request due to restrictions eg privacy or ethical. The data presented in this study are available on request from the corresponding author. The data are not publicly available due to privacy.

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
