# Peer review of "Evaluation of the Efficiency of Random and Diblock Methacrylate-Based Amphiphilic Cationic Polymers against Major Bacterial Pathogens Associated with Cystic Fibrosis"

_antibiotics, 2023, doi:10.3390/antibiotics12010120_

Round 1

Reviewer 1 Report

Dear Authors, 

The manuscript submitted for review contains absorbing research. The search for materials that are safe for human use and, at the same time, relatively simple to synthesize and have bactericidal functions is definitely highly recommended. Moreover, the application possibilities of such materials can be extremely wide. Therefore, it is worth looking into  chemical and electron structures of the polymers. 

After reading the manuscript, I divide the comments into two groups. The first group is about language issues. The second group is all other remarks with my questions or comments. 

1 Language comments:

(a) To improve the readability of selected sentences, You could add a comma (example: before the word 'and', after 'i.e.', before 'although'). Lines: 24, 32, 33, 98, 103, 138, 192, 220, 231, 243,244, 251, 265, 268, 268, 269, 282, 285, 285, 295, 313, 321, 340, 340, 354, 370, 390, 410, 421, 431, 466, 557;

(b) The use of proper prepositions is sometimes not so obvious. Nevertheless, You can consider replacing some of them ('on'>'against'; 'on'>'for'; 'with'>'for'; 'by'>'with'; 'against'>'to'). Lines: 24, 30, 37, 112, 348, 423, 150, 162, 188, 610, 378, 382;

(c) Before nouns or noun phrases, you can consider usnig of relevant articles ('a rapid'; 'a preventive'; 'the prevalence'; 'the life'; 'the bacterial'; 'a hydrophilic'; 'a hydrophobic'; 'the synthesis'; 'the presence'; 'an increase'; 'the higher'; 'the reasazurin'; 'a control'; 'a membranolytic'; 'the results'; 'a positive'; 'the extracted'; 'the bacterial'; 'the same'; 'a large'; 'a serial'; 'a viability'; 'a reconstructed'). Lines: 35, 38, 66, 69, 98, 120, 120, 130, 135, 163, 194, 203; 240, 249, 258, 279, 297, 303, 306, 313, 430, 519, 556, 607;

(d) I think the spelling of some words ('multidrug'; 'large-scale'; 'health care'; 'negatively charged'; 'round-bottom flask'; 'number average'; 'addition'; 're-extracted'; 're-suspended') can be rethought. Lines: 94, 115, 379, 401, 467, 475, 660.

(e) You might consider using a different form of the words derived from plural or grammatical tenses ('copolymers'; 'values'; 'activity'; 'enzymes'; 'lead'). Lines" 195, 210, 216, 221, 330, 407. 

(f) There are many long subordinate sentences in the manuscript. Most of them can be divided into smaller sentences. It will improve the readability of the entire manuscript.

2. Substantive remarks:

a) In my opinion, there is little correlation between the proposed title of the manuscript and the final conclusions. If youe retains such a title, one would definitely be tempted to go into more depth about the electrical charge generated on the polymer surface. The conclusions should at least include a brief summary of the discussion. 

b) The current version of the conclusions is too general and much too short.

c) The essential paragraph of the entire work (Lines 397-403) should be moved or paraphrased in the conclusions.

d) Bacteria are a group of microorganisms that are very diverse, and not included in a unified classification system. Bacteria do not have secreted intracellular structures. However, the most important structural element of the bacterial cell is a rigid cell wall. The authors have shown studies on both: Gram-positive and Gram-negative bacteria, (bacteria with different cell wall thicknesses). It is worth combining this information with the fact that the bactericidal activity of various materials is also associated with being a reducer. The DMAEMA monomer contains a primary amine prior to synthesis. The characteristic of these structures is that they have alkaline properties. Therefore, their aqueous solutions are alkaline. In the reaction thyp type of an amine with acids, the electron pair of nitrogen forms a coordination bond with the acid. It is also evident in the reaction notation for Diblock Copolymers, Figure 1. -NH3+ obtained as a result of protonation of the group-NH2 becomes a type II substituent or potential reducer. Under favorable conditions, by releasing electrons, it could destroy the bacteria cell wall. The free electron pair on nitrogen has a more share of the p orbital (orbital with higher energy). In principle, this involves the need to stabilize the surface charge. Summarizing, the more neutral the electric charge on the surface, the less active the compound should be. It is true that the presence of solvent is also essential and can change a lot. Nevertheless, it is worth checking the surface charge of the obtained polymers. Because of this, the synthesis will be more aware and may allow us to suggest why, among the synthesized polymers (differentiated mainly by their length), some are active and some not. The manuscript does not answer to this question.

I hope my comments will allow you to look at the presented results from a different poin of view. Thus, it will improve the presented results.

Sincerely yours, 

Rewiever

Author Response

Marseille, the 02th January 2023,

Dear Editor,

  On behalf of all coauthors and co-corresponding authors, I am submitting the revised version of our manuscript now entitled, following reviewers suggestions, Evaluation of the efficiency of Random and Diblock Methacrylate-based amphiphilic cationic polymers against major bacterial pathogens associated with Cystic Fibrosis (original title: Strengths and limitations of Random and Diblock Methacrylate-based amphiphilic cationic polymers against major bacterial pathogens associated with Cystic Fibrosis)

for consideration as a Research Article for publication in Antibiotics.

We sincerely thank the reviewers for constructive criticisms and valuable comments, which were of great help in revising the manuscript. Our responses to the reviewer’s comments are given below.

We hope our revised manuscript will be found suitable for publication in Antibiotics.

.

Sincerely yours,
Dr Marc Maresca

Reviewer 1 :

“Dear Authors, 

The manuscript submitted for review contains absorbing research. The search for materials that are safe for human use and, at the same time, relatively simple to synthesize and have bactericidal functions is definitely highly recommended. Moreover, the application possibilities of such materials can be extremely wide. Therefore, it is worth looking into  chemical and electron structures of the polymers. 

After reading the manuscript, I divide the comments into two groups. The first group is about language issues. The second group is all other remarks with my questions or comments.” 

Answer : We would like to thank the Reviewer for her/his supportive comments. Please find below our answers to his/her suggestions and questions.

“1 Language comments:

(a) To improve the readability of selected sentences, You could add a comma (example: before the word 'and', after 'i.e.', before 'although'). Lines: 24, 32, 33, 98, 103, 138, 192, 220, 231, 243,244, 251, 265, 268, 268, 269, 282, 285, 285, 295, 313, 321, 340, 340, 354, 370, 390, 410, 421, 431, 466, 557”

Answer : We would like to thank the Reviewer for her/his suggestions. Changes were done in the revised version of the manuscript.

“(b) The use of proper prepositions is sometimes not so obvious. Nevertheless, You can consider replacing some of them ('on'>'against'; 'on'>'for'; 'with'>'for'; 'by'>'with'; 'against'>'to'). Lines: 24, 30, 37, 112, 348, 423, 150, 162, 188, 610, 378, 382;”

Answer : We would like to thank the Reviewer for her/his suggestions. Manuscript has been checked and corrections were done when required.

“(c) Before nouns or noun phrases, you can consider usnig of relevant articles ('a rapid'; 'a preventive'; 'the prevalence'; 'the life'; 'the bacterial'; 'a hydrophilic'; 'a hydrophobic'; 'the synthesis'; 'the presence'; 'an increase'; 'the higher'; 'the reasazurin'; 'a control'; 'a membranolytic'; 'the results'; 'a positive'; 'the extracted'; 'the bacterial'; 'the same'; 'a large'; 'a serial'; 'a viability'; 'a reconstructed'). Lines: 35, 38, 66, 69, 98, 120, 120, 130, 135, 163, 194, 203; 240, 249, 258, 279, 297, 303, 306, 313, 430, 519, 556, 607;”

Answer : We would like to thank the Reviewer for her/his suggestions. Manuscript has been checked and corrections were done when required.

“(d) I think the spelling of some words ('multidrug'; 'large-scale'; 'health care'; 'negatively charged'; 'round-bottom flask'; 'number average'; 'addition'; 're-extracted'; 're-suspended') can be rethought. Lines: 94, 115, 379, 401, 467, 475, 660.”

Answer : We would like to thank the Reviewer for her/his suggestions. Manuscript has been checked and corrections were done when required.

“(e) You might consider using a different form of the words derived from plural or grammatical tenses ('copolymers'; 'values'; 'activity'; 'enzymes'; 'lead'). Lines" 195, 210, 216, 221, 330, 407. “

Answer : We would like to thank the Reviewer for her/his suggestions. Manuscript has been checked and corrections were done when required.

“(f) There are many long subordinate sentences in the manuscript. Most of them can be divided into smaller sentences. It will improve the readability of the entire manuscript.”

Answer : We would like to thank the Reviewer for her/his suggestions. Accordingly, we divided long sentences in shorter ones (Lines 62, 83, 279, 305, 310, 362, 398, 423).

“2. Substantive remarks:

  1. a) In my opinion, there is little correlation between the proposed title of the manuscript and the final conclusions. If you retains such a title, one would definitely be tempted to go into more depth about the electrical charge generated on the polymer surface. The conclusions should at least include a brief summary of the discussion.” 

Answer : We would like to thank the Reviewer for her/his suggestions. Accordingly, we changed the title to « Evaluation of the efficiency of Random and Diblock Methacrylate-based amphiphilic cationic polymers against major bacterial pathogens associated with Cystic Fibrosis ».

“b) The current version of the conclusions is too general and much too short. The essential paragraph of the entire work (Lines 397-403) should be moved or paraphrased in the conclusions.”

Answer : We would like to thank the Reviewer for her/his suggestions. Accordingly, we paraphrased the content of the discussion (lines 397-403) into the conclusions section (lines 678 to 692).

“c) Bacteria are a group of microorganisms that are very diverse, and not included in a unified classification system. Bacteria do not have secreted intracellular structures. However, the most important structural element of the bacterial cell is a rigid cell wall. The authors have shown studies on both: Gram-positive and Gram-negative bacteria, (bacteria with different cell wall thicknesses). It is worth combining this information with the fact that the bactericidal activity of various materials is also associated with being a reducer. The DMAEMA monomer contains a primary amine prior to synthesis. The characteristic of these structures is that they have alkaline properties. Therefore, their aqueous solutions are alkaline. In the reaction thyp type of an amine with acids, the electron pair of nitrogen forms a coordination bond with the acid. It is also evident in the reaction notation for Diblock Copolymers, Figure 1. -NH3+ obtained as a result of protonation of the group-NH2 becomes a type II substituent or potential reducer. Under favorable conditions, by releasing electrons, it could destroy the bacteria cell wall. The free electron pair on nitrogen has a more share of the p orbital (orbital with higher energy). In principle, this involves the need to stabilize the surface charge. Summarizing, the more neutral the electric charge on the surface, the less active the compound should be. It is true that the presence of solvent is also essential and can change a lot. Nevertheless, it is worth checking the surface charge of the obtained polymers. Because of this, the synthesis will be more aware and may allow us to suggest why, among the synthesized polymers (differentiated mainly by their length), some are active and some not. The manuscript does not answer to this question.”

Answer : We would like to thank the Reviewer for her/his suggestions, neverheless we would respectfully notice that all the copolymers have tertiary amine during the synthesis and not primary amine. Secondly the amine are quaternized with methyl iode to give them permanent cationic charge that is not dependant of the pH or conditions. In that case only the attractive forces between anionic bacterial membranes and the positiely charged copolymers occurred. There is no reduce functionality that could play a role as already highlithed in hundred of publication in this field. The occurrence of the charge in the copolymer is either totally random or located only on one block of the copolymer (diblock). In that case only the length (number of charge per chain), copolymer structure (random or diblock) and the FDMAEMA ratio (ratio between cationic and hydrophobic monomers) are parameters of interest. Such parameters have been screened into our copolymer library and the rationalization of the effects have been more cleary desbribed in the text.

Reviewer 2 Report

1- in lines 118 to 124, Here (aim of your work), you should only focus on your work and not include any references or other works.

2- in line 137 to 141, here, aim of the work, you start in the present study, .........

3- Tables 1 and 2 should be merged into a single table.

4- in line 145 to 147, You should write about your results in the results section without including any references or discussing other work.

5- table 1, 2 The data in the table should be represented in detail in the text, and  polymers and the greatest effect should be determined using the lowest value of mic.

6- line 157, remove it not necessary MIC vales of ........

7- You should statistically analyze your data and include it in the results section.

8- The data in all tables and figures should be represented in detail in the text, and  polymers and the greatest effect as IC50 separately

9- in line 435, You stated that you synthesized and characterized these polymers, and these tables are the results, which are written in the results section not in material and method section

10- time killing assay, add a reference

11- You should write about statistical analysis in the material and method section because you mentioned that you used three replicates in their experiments.

12- in line 673, rephrase it

13- You should double-check that any references listed in the text are correct.

14- in reference no 1, which is this reference, check

Author Response

Marseille, the 02th January 2023,

Dear Editor, Dear Reviewer,

  On behalf of all coauthors and co-corresponding authors, I am submitting the revised version of our manuscript now entitled, following reviewers suggestions, Evaluation of the efficiency of Random and Diblock Methacrylate-based amphiphilic cationic polymers against major bacterial pathogens associated with Cystic Fibrosis (original title: Strengths and limitations of Random and Diblock Methacrylate-based amphiphilic cationic polymers against major bacterial pathogens associated with Cystic Fibrosis)

for consideration as a Research Article for publication in Antibiotics.

We sincerely thank the reviewers for constructive criticisms and valuable comments, which were of great help in revising the manuscript. Our responses to the reviewer’s comments are given below.

We hope our revised manuscript will be found suitable for publication in Antibiotics.

.

Sincerely yours,
Dr Marc Maresca

1- in lines 118 to 124, Here (aim of your work), you should only focus on your work and not include any references or other works.

Answer : We would like to thank the Reviewer for her/his suggestions. Accordingly, the text was modified in the revised version of the manuscript and the reference was removed.

2- in line 137 to 141, here, aim of the work, you start in the present study, .........

Answer : We confirm that section from lines 118 to 143 describes the work.

3- Tables 1 and 2 should be merged into a single table.

Answer : We would like to thank the Reviewer for her/his suggestions. Accordingly, Tables 1 and 2 have been merged in the revised manuscript. Similarly, to be consistent, we also merged Tables 3 and 4 (new Table 2) and Tables 5 and 6 (new Table 3)

4- in line 145 to 147, You should write about your results in the results section without including any references or discussing other work.

Answer : We would like to thank the Reviewer for her/his suggestions. Accordingly, we removed this sentence and the reference associated from the revised manuscript.

5- table 1, 2 The data in the table should be represented in detail in the text, and polymers and the greatest effect should be determined using the lowest value of mic.

Answer : We would like to thank the Reviewer for her/his suggestions. Accordingly, more details were given in the text of the revised manuscript. We would like to indicate to the reviewer that the text clearly indicates that the more active copolymers are of course the ones with the lowest MIC values.

6- line 157, remove it not necessary MIC vales of ........

Answer : We would like to thank the Reviewer for her/his suggestions. Accordingly, text has been modified in the revised version.

7- You should statistically analyze your data and include it in the results section.

Answer : We would like to thank the Reviewer for her/his suggestions. Accordingly, comments have been added regarding the statistical significancy of the data (line 369). For other data (toxicity, killing, lipid insertion) dose or time-dependent effects were analyzed and plotted using GraphPad allowing to evaluate the significancy. Symbols of significancy in those figures have been omitted to avoid to make the figures very heavily loaded

8- The data in all tables and figures should be represented in detail in the text, and polymers and the greatest effect as IC50 separately

Answer : We would like to thank the Reviewer for her/his suggestions. Results are already described in more details (see answer to comment 5).

9- in line 435, You stated that you synthesized and characterized these polymers, and these tables are the results, which are written in the results section not in material and method section

Answer : We would like to thank the Reviewer for her/his suggestions. Accordingly, we moved previous tables 1 and 2 (new table 1) from the M&M and put it at the beginning of the results section.

10- time killing assay, add a reference

Answer : We would like to thank the Reviewer for her/his suggestions. A reference as been added.

11- You should write about statistical analysis in the material and method section because you mentioned that you used three replicates in their experiments.

Answer : We would like to thank the Reviewer for her/his suggestions. Text has been added in the revised version.

12- in line 673, rephrase it

Answer : We would like to thank the Reviewer for her/his suggestions. Sentence has been modified in the revised version.

13- You should double-check that any references listed in the text are correct.

Answer : We would like to thank the Reviewer for her/his suggestions. We have checked : references are correct.

14- in reference no 1, which is this reference, check

Answer : We would like to thank the Reviewer for her/his suggestions.We have changed this reference in the revised manuscript.

Reviewer 3 Report

This paper reported the methacrylate-based copolymers by radical chemistry, which did not induce resistance and remained active on antibiotic resistant strains, showing potential alternative to conventional antibiotics in the treatment of CF-associated bacterial infection.

 Specific comments

1.       I think the authors should add the detailed comparation to the conventional antibiotics.

2.       Some figures need to be revised, eg. Figures 4 and 5.

3.       The conclusions should be revised.

4.       The grammar for the whole manuscript should be revised carefully.

Author Response

Marseille, the 02th January 2023,

Dear Editor, Dear Reviewers,

  On behalf of all coauthors and co-corresponding authors, I am submitting the revised version of our manuscript now entitled, following reviewers suggestions, Evaluation of the efficiency of Random and Diblock Methacrylate-based amphiphilic cationic polymers against major bacterial pathogens associated with Cystic Fibrosis (original title: Strengths and limitations of Random and Diblock Methacrylate-based amphiphilic cationic polymers against major bacterial pathogens associated with Cystic Fibrosis)

for consideration as a Research Article for publication in Antibiotics.

We sincerely thank the reviewers for constructive criticisms and valuable comments, which were of great help in revising the manuscript. Our responses to the reviewer’s comments are given below.

We hope our revised manuscript will be found suitable for publication in Antibiotics.

.

Sincerely yours,
Dr Marc Maresca

Reviewer 1 :

“Dear Authors, 

The manuscript submitted for review contains absorbing research. The search for materials that are safe for human use and, at the same time, relatively simple to synthesize and have bactericidal functions is definitely highly recommended. Moreover, the application possibilities of such materials can be extremely wide. Therefore, it is worth looking into  chemical and electron structures of the polymers. 

After reading the manuscript, I divide the comments into two groups. The first group is about language issues. The second group is all other remarks with my questions or comments.” 

Answer : We would like to thank the Reviewer for her/his supportive comments. Please find below our answers to his/her suggestions and questions.

“1 Language comments:

(a) To improve the readability of selected sentences, You could add a comma (example: before the word 'and', after 'i.e.', before 'although'). Lines: 24, 32, 33, 98, 103, 138, 192, 220, 231, 243,244, 251, 265, 268, 268, 269, 282, 285, 285, 295, 313, 321, 340, 340, 354, 370, 390, 410, 421, 431, 466, 557”

Answer : We would like to thank the Reviewer for her/his suggestions. Changes were done in the revised version of the manuscript.

“(b) The use of proper prepositions is sometimes not so obvious. Nevertheless, You can consider replacing some of them ('on'>'against'; 'on'>'for'; 'with'>'for'; 'by'>'with'; 'against'>'to'). Lines: 24, 30, 37, 112, 348, 423, 150, 162, 188, 610, 378, 382;”

Answer : We would like to thank the Reviewer for her/his suggestions. Manuscript has been checked and corrections were done when required.

“(c) Before nouns or noun phrases, you can consider usnig of relevant articles ('a rapid'; 'a preventive'; 'the prevalence'; 'the life'; 'the bacterial'; 'a hydrophilic'; 'a hydrophobic'; 'the synthesis'; 'the presence'; 'an increase'; 'the higher'; 'the reasazurin'; 'a control'; 'a membranolytic'; 'the results'; 'a positive'; 'the extracted'; 'the bacterial'; 'the same'; 'a large'; 'a serial'; 'a viability'; 'a reconstructed'). Lines: 35, 38, 66, 69, 98, 120, 120, 130, 135, 163, 194, 203; 240, 249, 258, 279, 297, 303, 306, 313, 430, 519, 556, 607;”

Answer : We would like to thank the Reviewer for her/his suggestions. Manuscript has been checked and corrections were done when required.

“(d) I think the spelling of some words ('multidrug'; 'large-scale'; 'health care'; 'negatively charged'; 'round-bottom flask'; 'number average'; 'addition'; 're-extracted'; 're-suspended') can be rethought. Lines: 94, 115, 379, 401, 467, 475, 660.”

Answer : We would like to thank the Reviewer for her/his suggestions. Manuscript has been checked and corrections were done when required.

“(e) You might consider using a different form of the words derived from plural or grammatical tenses ('copolymers'; 'values'; 'activity'; 'enzymes'; 'lead'). Lines" 195, 210, 216, 221, 330, 407. “

Answer : We would like to thank the Reviewer for her/his suggestions. Manuscript has been checked and corrections were done when required.

“(f) There are many long subordinate sentences in the manuscript. Most of them can be divided into smaller sentences. It will improve the readability of the entire manuscript.”

Answer : We would like to thank the Reviewer for her/his suggestions. Accordingly, we divided long sentences in shorter ones (Lines 62, 83, 279, 305, 310, 362, 398, 423).

“2. Substantive remarks:

  1. a) In my opinion, there is little correlation between the proposed title of the manuscript and the final conclusions. If you retains such a title, one would definitely be tempted to go into more depth about the electrical charge generated on the polymer surface. The conclusions should at least include a brief summary of the discussion.” 

Answer : We would like to thank the Reviewer for her/his suggestions. Accordingly, we changed the title to « Evaluation of the efficiency of Random and Diblock Methacrylate-based amphiphilic cationic polymers against major bacterial pathogens associated with Cystic Fibrosis ».

“b) The current version of the conclusions is too general and much too short. The essential paragraph of the entire work (Lines 397-403) should be moved or paraphrased in the conclusions.”

Answer : We would like to thank the Reviewer for her/his suggestions. Accordingly, we paraphrased the content of the discussion (lines 397-403) into the conclusions section (lines 678 to 692).

“c) Bacteria are a group of microorganisms that are very diverse, and not included in a unified classification system. Bacteria do not have secreted intracellular structures. However, the most important structural element of the bacterial cell is a rigid cell wall. The authors have shown studies on both: Gram-positive and Gram-negative bacteria, (bacteria with different cell wall thicknesses). It is worth combining this information with the fact that the bactericidal activity of various materials is also associated with being a reducer. The DMAEMA monomer contains a primary amine prior to synthesis. The characteristic of these structures is that they have alkaline properties. Therefore, their aqueous solutions are alkaline. In the reaction thyp type of an amine with acids, the electron pair of nitrogen forms a coordination bond with the acid. It is also evident in the reaction notation for Diblock Copolymers, Figure 1. -NH3+ obtained as a result of protonation of the group-NH2 becomes a type II substituent or potential reducer. Under favorable conditions, by releasing electrons, it could destroy the bacteria cell wall. The free electron pair on nitrogen has a more share of the p orbital (orbital with higher energy). In principle, this involves the need to stabilize the surface charge. Summarizing, the more neutral the electric charge on the surface, the less active the compound should be. It is true that the presence of solvent is also essential and can change a lot. Nevertheless, it is worth checking the surface charge of the obtained polymers. Because of this, the synthesis will be more aware and may allow us to suggest why, among the synthesized polymers (differentiated mainly by their length), some are active and some not. The manuscript does not answer to this question.”

Answer : We would like to thank the Reviewer for her/his suggestions, neverheless we would respectfully notice that all the copolymers have tertiary amine during the synthesis and not primary amine. Secondly the amine are quaternized with methyl iode to give them permanent cationic charge that is not dependant of the pH or conditions. In that case only the attractive forces between anionic bacterial membranes and the positiely charged copolymers occurred. There is no reduce functionality that could play a role as already highlithed in hundred of publication in this field. The occurrence of the charge in the copolymer is either totally random or located only on one block of the copolymer (diblock). In that case only the length (number of charge per chain), copolymer structure (random or diblock) and the FDMAEMA ratio (ratio between cationic and hydrophobic monomers) are parameters of interest. Such parameters have been screened into our copolymer library and the rationalization of the effects have been more cleary desbribed in the text.

Reviewer 2 :

1- in lines 118 to 124, Here (aim of your work), you should only focus on your work and not include any references or other works.

Answer : We would like to thank the Reviewer for her/his suggestions. Accordingly, the text was modified in the revised version of the manuscript and the reference was removed.

2- in line 137 to 141, here, aim of the work, you start in the present study, .........

Answer : We confirm that section from lines 118 to 143 describes the work.

3- Tables 1 and 2 should be merged into a single table.

Answer : We would like to thank the Reviewer for her/his suggestions. Accordingly, Tables 1 and 2 have been merged in the revised manuscript. Similarly, to be consistent, we also merged Tables 3 and 4 (new Table 2) and Tables 5 and 6 (new Table 3)

4- in line 145 to 147, You should write about your results in the results section without including any references or discussing other work.

Answer : We would like to thank the Reviewer for her/his suggestions. Accordingly, we removed this sentence and the reference associated from the revised manuscript.

5- table 1, 2 The data in the table should be represented in detail in the text, and polymers and the greatest effect should be determined using the lowest value of mic.

Answer : We would like to thank the Reviewer for her/his suggestions. Accordingly, more details were given in the text of the revised manuscript. We would like to indicate to the reviewer that the text clearly indicates that the more active copolymers are of course the ones with the lowest MIC values.

6- line 157, remove it not necessary MIC vales of ........

Answer : We would like to thank the Reviewer for her/his suggestions. Accordingly, text has been modified in the revised version.

7- You should statistically analyze your data and include it in the results section.

Answer : We would like to thank the Reviewer for her/his suggestions. Accordingly, comments have been added regarding the statistical significancy of the data (line 369). For other data (toxicity, killing, lipid insertion) dose or time-dependent effects were analyzed and plotted using GraphPad allowing to evaluate the significancy. Symbols of significancy in those figures have been omitted to avoid to make the figures very heavily loaded

8- The data in all tables and figures should be represented in detail in the text, and polymers and the greatest effect as IC50 separately

Answer : We would like to thank the Reviewer for her/his suggestions. Results are already described in more details (see answer to comment 5).

9- in line 435, You stated that you synthesized and characterized these polymers, and these tables are the results, which are written in the results section not in material and method section

Answer : We would like to thank the Reviewer for her/his suggestions. Accordingly, we moved previous tables 1 and 2 (new table 1) from the M&M and put it at the beginning of the results section.

10- time killing assay, add a reference

Answer : We would like to thank the Reviewer for her/his suggestions. A reference as been added.

11- You should write about statistical analysis in the material and method section because you mentioned that you used three replicates in their experiments.

Answer : We would like to thank the Reviewer for her/his suggestions. Text has been added in the revised version.

12- in line 673, rephrase it

Answer : We would like to thank the Reviewer for her/his suggestions. Sentence has been modified in the revised version.

13- You should double-check that any references listed in the text are correct.

Answer : We would like to thank the Reviewer for her/his suggestions. We have checked : references are correct.

14- in reference no 1, which is this reference, check

Answer : We would like to thank the Reviewer for her/his suggestions.We have changed this reference in the revised manuscript.

Reviewer 3 :

“This paper reported the methacrylate-based copolymers by radical chemistry, which did not induce resistance and remained active on antibiotic resistant strains, showing potential alternative to conventional antibiotics in the treatment of CF-associated bacterial infection.

 Specific comments

  1. I think the authors should add the detailed comparation to the conventional antibiotics.
  2. Some figures need to be revised, eg. Figures 4 and 5.
  3. The conclusions should be revised.
  4. The grammar for the whole manuscript should be revised carefully.”

Answer : We would like to thank the Reviewer for her/his suggestions. We respectfully believe that our answers to comments from Reviewers 1 and 2 and the associated modifications of the text in the revised version will satisfy Reviewer 3. If it is not the case, we will respectfully ask Reviewer 3 to give us more details in his/her comments in order to allow us to perform required modifications.

Round 2

Reviewer 3 Report

The authors have revised the manuscript as required.